# Differentially Private Sequential Data Synthesis with Structured State Space Models and Diffusion Models

**Tomoya Matsumoto**[1]    **Takayuki Miura**[1,2]    **Toshiki Shibahara**[2]    **Masanobu Kii**[2]
**Kazuki Iwahana**[1,2]    **Osamu Saisho**[2]    **Shingo Okamura**[1,3]
[1]Osaka University    [2]NTT Social Informatics Laboratories
[3]National Institute of Technology, Nara College
t-matsumoto@ist.osaka-u.ac.jp
{tkyk.miura,toshiki.shibahara,masanobu.kii}@ntt.com
{kazuki.iwahana,osamu.saisho}@ntt.com
okamura@info.nara-k.ac.jp

## Abstract

Sequential data such as electrocardiograms and electroencephalograms are being increasingly utilized, and protecting the privacy of individuals in data has become an important issue. For statistical analysis while preserving privacy, data synthesis with differential privacy (DP) has been attracting attention. However, DP synthetic data generally suffers from a decrease in quality. In this paper, we aim to achieve high-quality DP synthesis for sequential data. First, we show that previous DP sequential data synthesis has quality issues. We then propose DP structured state space diffusion (DP-SSSD), a DP sequential data synthesis method based on novel generative AI, which combines structured state space models and diffusion models. Experiments show that DP-SSSD can generate higher-quality sequential data than the previous methods under equal privacy protection strength.

## 1   Introduction

With the spread of wearable devices and advances in data analysis technology, sequential data such as electrocardiograms (ECGs) and electroencephalograms (EEGs) are being increasingly collected and utilized [17, 4]. These data are valuable, for example, in the healthcare field, where they can be used for early detection and prevention of diseases. However, handling such data requires careful consideration because they contain private information such as personal health and should not be shared with third parties as they are.

To solve this issue, data synthesis techniques based on differential privacy (DP) [7] have attracted attention for statistical analysis of sensitive data. Its generated data are called privacy-preserving synthetic data because they retain the statistical characteristics of the raw data while making it difficult to invade the privacy of individuals [28]. However, it requires noise to be added to satisfy DP and thus suffers from a trade-off between utility and privacy protection strength. In particular, sequential data synthesis is a difficult task in itself, so improving the privacy-utility trade-off of differentially private sequential data is a major challenge.

Therefore, this paper aims to generate high-quality sequential data that satisfy DP by combining a structured state space model [13, 12] and a diffusion model [14], which are recent breakthroughs in the field of deep learning. The main contributions of this paper are as follows: (1) We demonstrate that previous differentially private sequential data synthesis [29] has quality issues. (2) We propose a differentially private generative model using structured state space models and diffusion models and evaluate its synthesis quality on ECG and EEG datasets. (3) We show that the proposed method can generate higher-quality sequential data than the previous method under equal privacy protection

NeurIPS 2024 Safe Generative AI Workshop.

strength. (4) To improve the privacy-utility trade-off, we test a method to reduce the amount of noise per parameter by fixing weight parameters that are not to be updated much during training and show its effectiveness in a classification task.

## 2 Preliminary

In this section, we introduce basic concepts and models to describe our results.

### 2.1 Differential Privacy (DP)

DP is a framework that guarantees the privacy of individuals in a dataset.

**Definition 2.1 (Neighboring datasets)** *Datasets $D, D' \in \mathcal{D}$ are called neighboring datasets if $D$ and $D'$ differ only in one record.*

**Definition 2.2 (($\epsilon, \delta$)-DP)** *A randomized function $f : \mathcal{D} \to \mathcal{R}$ satisfies ($\epsilon, \delta$)-DP if for any neighboring $D, D' \in \mathcal{D}$ and any range $S \subset \mathcal{R}$, it holds that*

$$\Pr[f(D) \in S] \le e^\epsilon \Pr[f(D') \in S] + \delta. \tag{1}$$

The smaller the values of $\epsilon, \delta$, the stronger the privacy protection. $\epsilon$ is called the privacy loss.

DP-SGD is a modified version of stochastic gradient descent (SGD) and is widely used to train neural networks with DP. In regular SGD, the weight parameters $\theta$ are updated using the gradient $\mathbf{g}$. In DP-SGD, the gradient is clipped, as shown in Eq. (2), and Gaussian noise is added to it. Here, $x_i$ is a data sample, $C$ is an upper bound of the norm, $\eta$ is the learning rate, $L$ is the batch size and $\sigma$ is the noise multiplier.

$$\bar{\mathbf{g}}_t(x_i) \leftarrow \mathbf{g}_t(x_i) / \max\left(1, \frac{\|\mathbf{g}_t(x_i)\|_2}{C}\right), \tag{2}$$

$$\theta_{t+1} \leftarrow \theta_t - \frac{\eta_t}{L}\left(\sum_i \bar{\mathbf{g}}_t(x_i) + \mathcal{N}(0, \sigma^2 C^2 \mathbf{I})\right). \tag{3}$$

The privacy loss in DP-SGD is calculated on the basis of the number of uses of the original dataset by the trained model. Many versions of DP-SGD have been proposed [1, 23, 11].

### 2.2 Structured State Space Model

The structured state space model is a sequential model that uses the state space representation expressed by Eq. (4). Here, $u(t) \in \mathbb{R}$ is the input signal, $x(t) \in \mathbb{R}^N$ is the latent state and $y(t) \in \mathbb{R}$ is the output signal, where $\mathbb{R}$ is the set of real numbers. $\boldsymbol{A} \in \mathbb{C}^{N \times N}, \boldsymbol{B} \in \mathbb{C}^{N \times 1}, \boldsymbol{C} \in \mathbb{C}^{1 \times N}$, and $\boldsymbol{D} \in \mathbb{R}$ are trainable parameters, where $\mathbb{C}$ is the set of complex numbers.

$$x'(t) = \boldsymbol{A}x(t) + \boldsymbol{B}u(t), \quad y(t) = \boldsymbol{C}x(t) + \boldsymbol{D}u(t). \tag{4}$$

By discretizing Eq. (4), it can be expressed in recursive form as in Eq. (5).[1] Here, $\Delta \in \mathbb{R}$ is a trainable parameter, which expresses the step size.

$$x_k = \overline{\boldsymbol{A}}x_{k-1} + \overline{\boldsymbol{B}}u_k, \quad y_k = \overline{\boldsymbol{C}}x_k,$$
$$\overline{\boldsymbol{A}} = \exp(\Delta\boldsymbol{A}), \quad \overline{\boldsymbol{B}} = (\Delta\boldsymbol{A})^{-1}(\exp(\Delta \cdot \boldsymbol{A}) - \boldsymbol{I}) \cdot \Delta\boldsymbol{B}, \quad \overline{\boldsymbol{C}} = \boldsymbol{C}. \tag{5}$$

Also, Eq. (5) can be expressed in convolution form as in Eq. (6).

$$y = u * \overline{\boldsymbol{K}}, \quad \overline{\boldsymbol{K}} = (\overline{\boldsymbol{CB}}, \overline{\boldsymbol{CAB}}, \dots, \overline{\boldsymbol{CA}}^{L-1}\overline{\boldsymbol{B}}). \tag{6}$$

The parameter $\boldsymbol{A}$ requires appropriate initial values. The parameter of S4D [12], which is a structured state space model, is expressed by the following diagonal matrix:

$$\boldsymbol{A}_{nn} = -\frac{1}{2} + i\pi n. \tag{7}$$

---

[1]The term $\boldsymbol{D}u$ is omitted because it is regarded as a skip connection.

## 2.3 Diffusion Model

A diffusion model [14] is a generative model consisting of a diffusion process and a denoising process described as follows;

$$q(x_t|x_{t-1}) = \mathcal{N}(x_t; \sqrt{1-\beta_t}x_{t-1}, \beta_t I), \quad p_\theta(x_{t-1}|x_t) = \mathcal{N}(x_{t-1}; \mu_\theta(x_t, t), \Sigma_\theta(x_t, t)). \quad (8)$$

In the diffusion process, we transform observed data $x_0$ into noise $x_T$ by adding Gaussian noise in accordance with the noise schedule $\beta_t$. In the denoising process, we reconstruct the data $x_0$ from the noise $x_T$ by repeating the denoising noises with a neural network $\theta$. The loss function is defined as follows:

$$L := \mathbb{E}_{t,x_0,\epsilon}\left[||\epsilon - \epsilon_\theta(\sqrt{\bar{\alpha}_t}x_0 + \sqrt{1-\bar{\alpha}_t}\epsilon, t)||^2\right]. \quad (9)$$

This expresses the error of the estimation of noise $\epsilon$. Here, we set $\bar{\alpha}_t := \prod_{s=1}^{t}(1-\beta_s)$. In the diffusion model, the neural network is trained to minimize this error.

## 3 Related Work

In this section, we describe sequential data synthesis and differentially private data synthesis as the related work.

### 3.1 Sequential Data Synthesis

The sequential data synthesis is generally performed using deep learning models, which can be further classified into four types depending on the generative model used: generative adversarial network (GAN), auto-regressive model, variational autoencoder (VAE), and diffusion model.

GAN has been the most widely used model in data synthesis. In particular, TimeGAN [33] is a well-established method for sequential data synthesis. However, when we experimented with ECG and EEG data, TimeGAN failed to generate useful samples. This has been pointed out in previous studies [2].

For the sequential data synthesis by auto-regressive models, the conditional probabilistic auto-regressive model (CPAR) [35] is widely used. CPAR is employed in the synthetic data library Synthetic Data Vault [26], but in our experiments, CPAR was unable to generate useful samples, similar to TimeGAN. Nishikimi et al. [25] proposed an approach for synthesizing sequential data using a VAE. This method uses ECG data on hearts simulated by a supercomputer, whereas we focus on only raw ECG data in this paper.

Diffusion models have recently garnered attention as data synthesis methods surpassing GANs. Alcaraz and Strodthoff [2] presented structured state space diffusion (SSSD), which synthesizes sequential data using a diffusion model. SSSD combines DiffWave [20], which is a diffusion-based speech synthesis method, with S4 [13], which is a type of structured state space model. Our experimental results on ECG and EEG data show that SSSD can generate higher-quality sequential data than conventional models. Therefore, we implement and evaluate a DP-compliant synthesis method based on SSSD in this paper.

### 3.2 Differentially Private Data Synthesis

The related works mentioned in Section 3.1 are all methods that do not satisfy DP. Therefore, in this section, we review related research on data synthesis methods that satisfy DP.

In our experiments, we use diffusion models and structured state space models, and one diffusion model that satisfies DP is Differentially Private Diffusion Model (DPDM) [6]. DPDM trains a diffusion model using DP-SGD and introduces a technique called noise multiplicity to improve generation quality. However, DPDM is designed for image generation models and does not evaluate diffusion models that satisfy DP for sequential data synthesis. Additionally, to our knowledge, there is no research applying DP to structured state space models.

On the other hand, several studies provide differentially private sequential data synthesis [8, 3, 29]. All of these methods involve training GAN models with DP-SGD. In particular, Rényi DP and Convolutional GAN (RDP-CGAN) [29] focuses on synthesizing ECG and EEG data, the same

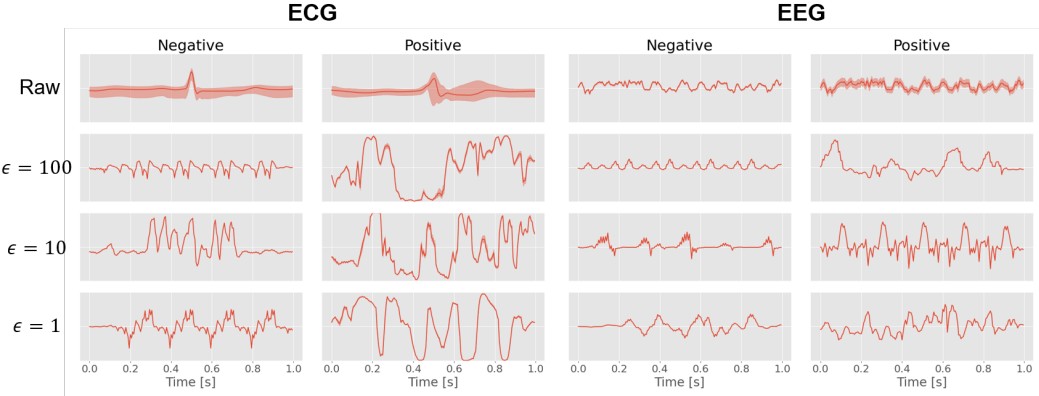

Figure 1: The first row is a waveform of raw data, and the second to fourth rows are waveforms generated by RDP-CGAN with varying epsilon values ($\epsilon = 100, 10, 1$). The first and second columns represent ECG, and the third and fourth columns represent EEG, each containing normal (negative) and abnormal (positive) samples, respectively.

as in this paper, so we evaluate RDP-CGAN as a baseline on the basis of the public source code. RDP-CGAN is a deep learning model that combines a one-dimensional convolutional neural network (1D-CNN) and GAN and adopts a method based on Rényi DP [22] to calculate privacy loss [23].

## 4 Evaluation of Previous Method

This section presents the issue of a previous method for differential private sequential data synthesis and describes the problem setup for the experiments in Section 5.

### 4.1 Issues in Previous Method

Existing studies of differentially private sequential data synthesis described in Section 3.2 face a major challenge in term of the quality of synthetic data. Figure 1 shows ECG and EEG waveforms generated by RDP-CGAN [29]. The figure shows that differentially private synthetic data struggles to reproduce the features and diversity of the raw data, regardless of privacy protection strength. This is largely due to a problem common to GAN models called mode collapse [10], which is particularly likely to occur in learning with DP-SGD. This is because DP-SGD cannot use batch normalization [16], which helps stabilize the learning of the GAN model, to limit the impact of each sample on the sum of gradients. Instead, group normalization [32] is used, but this change prevents cross-sample normalization, making the GAN model training unstable and prone to mode collapse. Therefore, a generative model needs to be used that is less prone to mode collapse to generate high-quality differentially private synthetic data.

### 4.2 Purpose of This Paper

Considering the issues of the previous study, this paper proposes a differentially private generative model based on SSSD [2], a synthesis method using a diffusion model. Specifically, we applied DP-SGD to the SSSD model that is not differentially private to make it satisfy DP. This differentially private SSSD model is hereafter referred to as DP-SSSD.

Section 5 proceeds as follows. Section 5.2.1 evaluates and compares the synthesis quality of the structured state space model with that of other sequential models. The SSSD model uses structured state space models for the inner sequential model, and an experiment to verify whether the choice is optimal is described. Section 5.2.2 evaluates and compares the synthesis quality of DP-SSSD with that of the previous method, RDP-CGAN. The purpose of this experiment was to clarify whether DP-SSSD can synthesize higher-quality sequential data than RDP-CGAN.

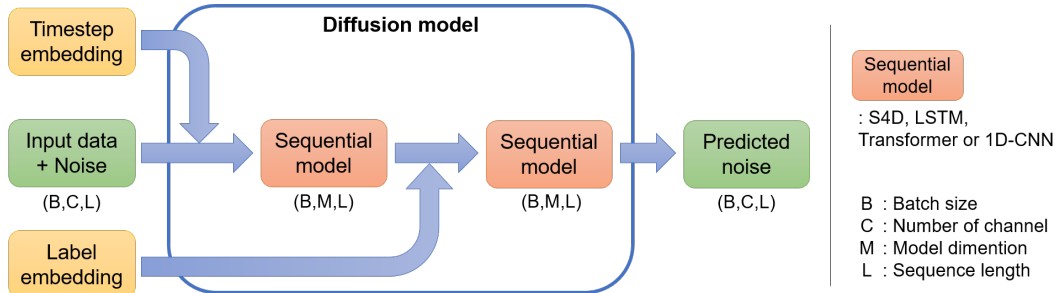

Figure 2: Overview of the implemented model. DP-SSSD is the one that uses S4D for the sequential model.

# 5    Experiments

In this section, we evaluate the synthesis performance of the proposed method on ECG and EEG data.

## 5.1    Experimental Settings

### 5.1.1    Datasets

We use ECG and EEG datasets in the experiments because they are widely used in sequential data synthesis research and are considered to be of high privacy-preserving importance. We use the MIT-BIH arrhythmia database [24, 9] for ECG and the epileptic seizure recognition dataset from the UCI Machine Learning Repository [19] for EEG. Hereafter, we refer to them as ECG/EEG data.

Both datasets are preprocessed and used as single-channel signals with a sampling frequency of 180 Hz for 1 second. The ECG samples are labeled as normal or ventricular ectopic beats, and the EEG samples are labeled as normal or epileptic seizures. In this paper, normal samples are labeled negative, and abnormal samples are labeled positive. Each dataset is divided into training and test data. The training data is used to train the generative model and the test data is used to evaluate the performance of the classification models, as described below. In the experiments in this paper, the number of samples generated by a generative model is always equal to the number of samples of training data. Table 1 shows a breakdown of the number of samples for each dataset.

### 5.1.2    Synthesis Method

DP-SSSD uses a structured state space model as a sequential model, with the model overview shown in Figure 2. The original SSSD model uses a structured state space model called S4 [13], but DP-SSSD uses S4D [12], which is smaller than S4 and has the same level of performance. The generative model is smaller than the original SSSD model because the smaller the number of parameters in the model, the smaller the effect of noise added by DP-SGD [6].

We used the Python library Opacus [34] to implement DP-SGD and the privacy loss was calculated using the library's standard method, Privacy Loss Random Variables (PRVs) [11]. We also reimplemented RDP-CGAN, which was used as a prior method, using PRVs to ensure a fair comparison with DP-SSSD. The value of $\delta$, which affects the strength of privacy protection, was set to $5.0 \times 10^{-5}$ for the ECG data and $1.0 \times 10^{-4}$ for the EEG data on the basis of the number of samples.

### 5.1.3    Evaluation Metrics

As reliable evaluation metrics based on the results of previous studies and preliminary experiments, we adopted the following three metrics: (1) For visual evaluation, we plot the waveform of synthetic sequential data. The more similar the shape is to that of the raw data, the higher the synthesis quality. Following previous studies [2], we draw the median of all samples as a solid line and the interquartile range as a band. (2) We compare the performances of classification models trained on the labeled synthetic data. The smaller the drop in the performance when trained on the raw data, the higher the synthesis quality. We use CatBoost [27], a decision tree-based model, and a deep learning-based model (Deep Neural Network; DNN) [18]. Following the previous study [29], the area

Table 1: The left side shows the number of samples and positive rates for each dataset, while the right side shows the performance of classification models trained on the raw data.

| Dataset | Training data | | Test data | | AUROC | | AUPRC | |
| --- | --- | --- | --- | --- | --- | --- | --- | --- |
| | Num. of samples | Pos. rate | Num. of samples | Pos. rate | CatBoost | DNN | CatBoost | DNN |
| ECG | 30,810 | 20% | 24,179 | 3.2% | 0.81 ±.007 | 0.87 ±.009 | 0.45 ±.035 | 0.23 ±.060 |
| EEG | 9,200 | 20% | 2,300 | 20% | 1.00 ±.000 | 0.99 ±.001 | 0.98 ±.001 | 0.95 ±.003 |

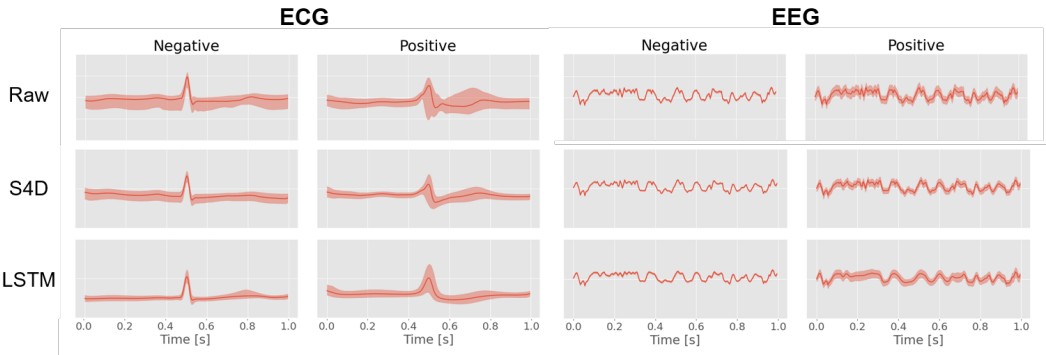

Figure 3: The first row is a waveform of raw ECG/EEG, and the second and third rows are waveforms of synthetic ECG/EEG using S4D/LSTM for the sequential model.

Table 2: Classification performance trained on synthetic ECG/EEG using S4D/LSTM for the sequential model.

| Dataset | Sequential model | AUROC | | AUPRC | |
| --- | --- | --- | --- | --- | --- |
| | | CatBoost | DNN | CatBoost | DNN |
| ECG | S4D | 0.75 ±.002 | 0.82 ±.009 | **0.56 ±.037** | 0.37 ±.065 |
| | LSTM | 0.80 ±.009 | **0.88 ±.024** | 0.27 ±.053 | 0.19 ±.018 |
| EEG | S4D | 0.98 ±.001 | 0.98 ±.002 | 0.93 ±.012 | **0.94 ±.009** |
| | LSTM | **0.99 ±.000** | 0.98 ±.001 | 0.92 ±.005 | **0.94 ±.003** |

under the receiver operating characteristic curve (AUROC) and the area under the precision-recall curve (AUPRC) are used as the performance metrics. Table 1 shows the performance of classification models trained on the raw training data. (3) We compare the distribution of synthetic data using the dimensionality reduction technique. The closer the distribution is to that of the raw data, the higher the synthesis quality. We use t-distributed Stochastic Neighbor Embedding (t-SNE) [30] as the dimensionality reduction technique.

## 5.2 Results

### 5.2.1 Comparison of Structured State Space Models with Other Sequential Models

First, to clarify whether the structured state space model (S4D) is the most appropriate sequential model to be included in the diffusion model, we compared the synthesis quality using S4D with other sequential models. For comparison, we chose LSTM [15], Transformer [31] and 1D-CNN, which are representative sequential models. However, Transformer and 1D-CNN were unable to generate meaningful samples, so only the results for S4D and LSTM are included in this paper.

Figure 3 shows the synthetic ECG and EEG using S4D or LSTM. This figure shows that S4D can reproduce the detailed features of the raw data waveforms better than LSTM. The difference is especially noticeable in the positive EEG samples. Table 2 shows the performance of classification models trained on each synthetic data. S4D has a higher AUPRC on ECG data than LSTM, which means the model can be trained to correctly classify a small number of positive samples. In the visualized data distribution shown in Figure 5, the synthetic data distribution using S4D is closer to the raw data distribution than that using LSTM is. These results indicate that the structured state space model (S4D) is suitable for sequential data synthesis.

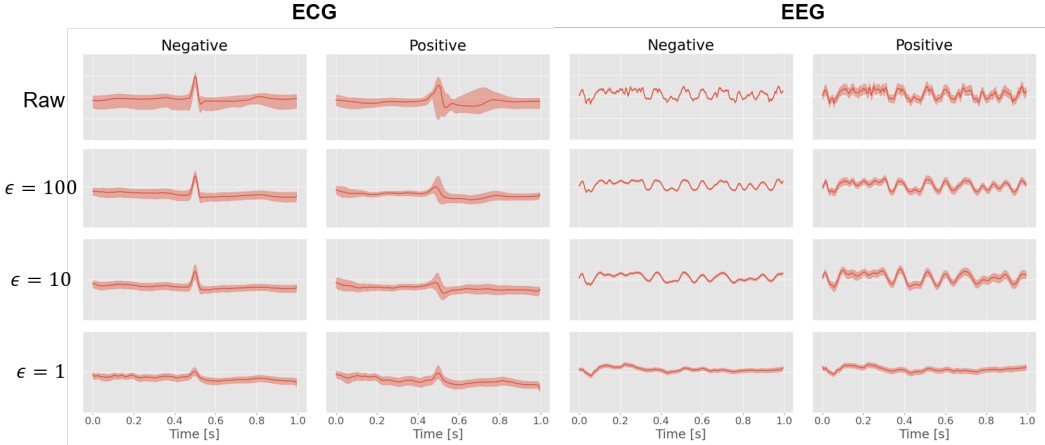

Figure 4: The first row is a waveform of raw ECG/EEG, and the second to fourth rows are waveforms generated by DP-SSSD with varying epsilon values ($\epsilon = 100, 10, 1$).

Table 3: Classification performance trained on differentially private synthetic ECG/EEG generated by DP-SSSD/RDP-CGAN.

| Method | $\epsilon$ | ECG | | | | EEG | | | |
|---|---|---|---|---|---|---|---|---|---|
| | | AUROC | | AUPRC | | AUROC | | AUPRC | |
| | | CatBoost | DNN | CatBoost | DNN | CatBoost | DNN | CatBoost | DNN |
| DP-SSSD | 100 | 0.81 ±.034 | **0.86 ±.024** | **0.52 ±.014** | 0.33 ±.038 | **0.98 ±.000** | **0.98 ±.003** | 0.92 ±.010 | **0.94 ±.000** |
| | 10 | 0.77 ±.006 | **0.85 ±.028** | **0.45 ±.002** | 0.16 ±.008 | **0.98 ±.000** | 0.97 ±.014 | 0.89 ±.000 | **0.93 ±.017** |
| | 1 | **0.86 ±.002** | 0.62 ±.017 | **0.16 ±.006** | 0.05 ±.000 | **0.97 ±.002** | **0.97 ±.003** | 0.91 ±.010 | **0.93 ±.003** |
| RDP-CGAN | 100 | 0.64 ±.049 | 0.53 ±.059 | 0.05 ±.009 | 0.05 ±.020 | 0.84 ±.001 | 0.72 ±.315 | 0.74 ±.020 | 0.62 ±.354 |
| | 10 | 0.50 ±.061 | 0.51 ±.023 | 0.04 ±.008 | 0.04 ±.004 | 0.74 ±.097 | 0.89 ±.060 | 0.54 ±.123 | 0.84 ±.055 |
| | 1 | 0.66 ±.015 | 0.54 ±.165 | 0.05 ±.001 | 0.07 ±.057 | 0.81 ±.000 | 0.85 ±.054 | 0.68 ±.000 | 0.77 ±.024 |

### 5.2.2 Comparison of DP-SSSD with RDP-CGAN

We evaluated the synthesis quality of DP-SSSD and RDP-CGAN under various privacy loss values $\epsilon$. Figure 4 shows the ECG and EEG waveforms generated by DP-SSSD. Comparing this figure with the waveforms in Figure 1, DP-SSSD can clearly generate differentially private synthetic data more similar to the raw data waveforms than RDP-CGAN. DP-SSSD succeeds in generating waveforms that satisfy DP without mode collapse due to stable learning, which is one of the advantages of the diffusion model. However, the stronger the privacy protection, the more features of the raw data are lost, and especially for $\epsilon = 1$, it fails to generate meaningful waveforms.

Table 3 shows the performance of classification models trained on differentially private synthetic ECG/EEG. DP-SSSD outperformed RDP-CGAN for all $\epsilon$ values. Figure 6 shows the distribution of differentially private synthetic data visualized by t-SNE. This figure also shows that DP-SSSD can generate more diverse data than RDP-CGAN. These results indicate that DP-SSSD can generate differentially private sequential data of higher quality than RDP-CGAN in the previous study. On the other hand, in a strong privacy protection setting such as $\epsilon = 1$, waveforms that resemble the raw data are difficult to generate even with DP-SSSD.

## 6 Additional Analysis

In this section, we propose a method to improve the privacy-utility trade-off of structured state space models and examine its effectiveness.

### 6.1 Fixing Weight Parameters

The experimental results in the previous section revealed that the quality of synthetic waveforms deteriorates significantly when privacy protection is strengthened. Therefore, it is necessary to reduce the effect of noise added to the gradient in DP-SGD and improve the quality of the learning

Table 4: S4D classification accuracy on sCIFAR dataset. The left and right columns show the cases where the parameters are not fixed and are fixed, respectively.

| $\epsilon$ | No fix | Fix |
|---|---|---|
| inf | **76.2 ±0.18** | 75.1 ±0.20 |
| 100 | 53.0 ±0.47 | **55.2 ±0.93** |
| 10 | 48.8 ±0.36 | **50.7 ±0.21** |
| 1 | 42.3 ±0.65 | **44.0 ±0.30** |

Table 5: Classification performance trained on synthetic ECG/EEG by DP-SSSD when the weights are fixed and the difference in performance when the weights are not fixed. △ represents a performance improvement and ▼ represents a deterioration.

| Dataset | $\epsilon$ | AUROC | | AUPRC | |
|---|---|---|---|---|---|
| | | CatBoost | DNN | CatBoost | DNN |
| ECG | 100 | 0.79 (▼.02) | 0.89 (△.03) | 0.56 (△.04) | 0.45 (△.12) |
| | 10 | 0.84 (△.07) | 0.85 (-.00) | 0.53 (△.08) | 0.23 (△.07) |
| | 1 | 0.72 (▼.14) | 0.63 (△.01) | 0.19 (△.03) | 0.05 (-.00) |
| EEG | 100 | 0.98 (-.00) | 0.98 (-.00) | 0.88 (▼.04) | 0.94 (-.00) |
| | 10 | 0.98 (-.00) | 0.98 (△.01) | 0.92 (△.03) | 0.94 (△.01) |
| | 1 | 0.95 (▼.02) | 0.95 (▼.02) | 0.86 (▼.05) | 0.90 (▼.03) |

results. While some DP-SGD learning methods have been proposed for diffusion models and transformers [6, 5], this section examines the effectiveness of a simple method focused on structured state space models (S4D).

Specifically, differentially private learning is performed with $A$ and $\Delta$, two of the weight parameters of the structured state space model described in Section 2.2, fixed at their initial values. $A$ and $\Delta$ are parameters that repeatedly affect the latent state $x$ in Equation (5), and since small changes in these parameters greatly impact the model, a small learning rate is specially set. Preliminary experiments have shown that the usual non-DP learning with these parameters fixed to their initial values barely affects the accuracy. As mentioned in Section 5.1.2, the smaller the number of model parameters in DP-SGD, the smaller the effect of added noise, so we will experimentally verify whether fixing the above parameters improves the quality of differentially private learning.

## 6.2 Effectiveness of Parameter Fixing

First, we examine the effect of parameter fixing on a classification task using the structured state space model (S4D) alone. The reason for this is that there are few weight parameters other than the S4D model, and the effect of parameter fixing is easy to understand. A classifier using the S4D model is trained on the flattened CIFAR-10 [21] (sCIFAR) [12]. Table 4 shows the classification accuracy when parameters are and are not fixed. The classification accuracy is higher when the parameters are not fixed in the usual non-DP training, but the accuracy is higher when the parameters are fixed in differentially private training.

Next, the effect of fixing parameters is verified by generating ECG and EEG using the same DP-SSSD as in Section 5. Table 5 shows the performance of classification models trained on differentially private synthetic data when the parameters are fixed. Figure 7 shows the differentially private synthetic (positive) waveforms with and without parameter fixing. When the parameters were fixed, although the quality of the synthetic waveforms did not improved, the AUPRC of the ECG did. Thus, parameter fixing can be said to effectively improve some tasks.

## 7 Conclusion

In this paper, we proposed differential privacy structured state space diffusion (DP-SSSD), a differentially private synthesis method combining structured state space models and diffusion models to achieve high-quality sequential data synthesis that satisfies DP. We evaluated it using ECG and EEG datasets and experimental results showed that DP-SSSD can generate higher-quality sequential data than previous methods under equal privacy protection strength, as indicated by several evaluation metrics. In addition, we verified that fixing some of the weight parameters is effective to improve quality, especially in the classification task.

Future work is to develop a method that can generate synthetic sequences that accurately capture the characteristics of the raw data even in strong privacy-preserving settings. This will require a sequential generative model that is more compact and robust to noise. Future work includes validation using long-time sequential data and sequences other than biosignals.

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

# A The Figures of Experimental Results in Section 5

We show the visualized data distribution of the experiment in Section 5.

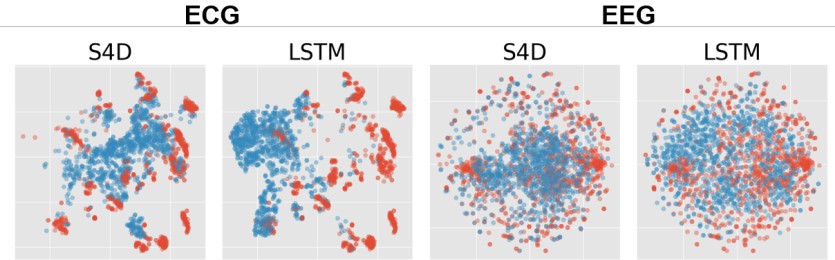

Figure 5: Distribution of synthetic ECG/EEG visualized by t-SNE (red: raw data, blue: synthetic data). The closer the two distributions are, the higher the quality of the synthetic data.

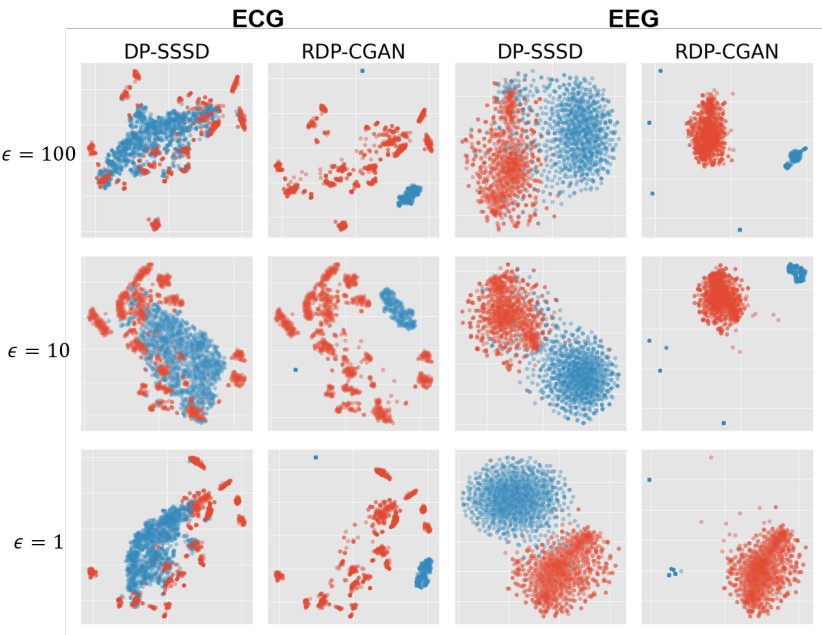

Figure 6: Distribution of differentially private synthetic ECG/EEG visualized by t-SNE (red: raw data, blue: synthetic data).

# B The Figures of Experimental Results in Section 6

We show the ECG/EEG waveforms of the experiment in Section 6.

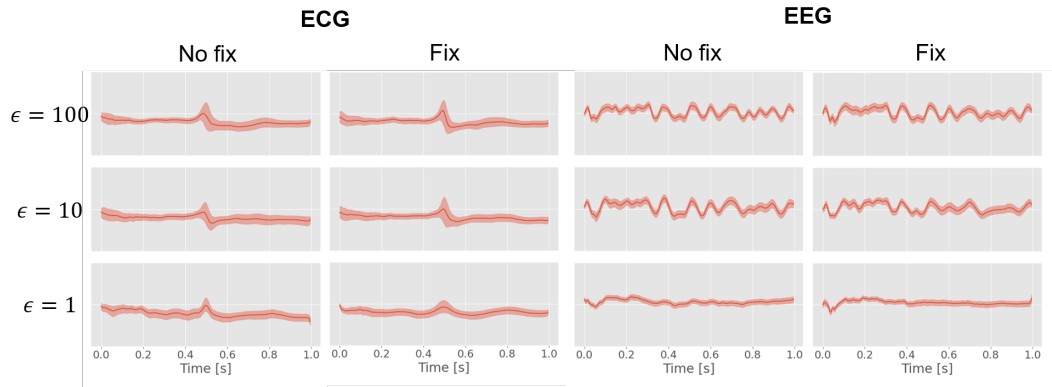

Figure 7: ECG/EEG waveforms generated by DP-SSSD with and without fixing weight parameters.

