# OpenReview forum: "Differentially Private Sequential Data Synthesis with Structured State Space Models and Diffusion Models"
_NeurIPS.cc/2024/Workshop/SafeGenAi — SafeGenAi Poster_

### Official Review · Reviewer_2zP8 · 2024-10-09
**Unclear algorithm details**

**Rating:** 4
**Confidence:** 4

**Review:**

The authors propose a differentially private sequential data synthesis based on structured state space models with a diffusion model. Results on ECG and EEG data show some advantages compared with one previous method. However, several major concerns are listed below:

1. The algorithm details are not clearly explained, showing less confidence in the experimental results.

2. Some descriptions are not accurate. For example, the sentence on page 5 "The value of $\delta$, which affects the strength of privacy protection" is inappropriate. In fact, the added noise in Eq. (3) affects the privacy level. In addition, how to calculate the estimation $\epsilon$ is also omitted.

3. Abalation studies on the sequential model and other generative models are not complete. Only one model for each comparison is only, which shows limited information.

---

### Official Review · Reviewer_4HLF · 2024-10-09
**Dataset Proportions, Generalization on applications and Privacy Parameters guidance**

**Rating:** 5
**Confidence:** 5

**Review:**

Firstly, I don't think this paper is related to the AI safety concern topic of this workshop SafeGenAI.

My review:
This paper addresses an important problem in data privacy: generating high-quality differentially private (DP) synthetic sequential data, such as electrocardiograms (ECGs) and electroencephalograms (EEGs), using novel generative models. The authors propose a method called DP Structured State Space Diffusion (DP-SSSD), which combines structured state space models (S4D) with diffusion models to improve the quality of DP synthetic data. Through empirical evaluation on ECG and EEG datasets, the paper demonstrates that DP-SSSD outperforms previous methods, such as RDP-CGAN, in generating more realistic data under equivalent privacy constraints. Additionally, the authors explore techniques to improve the privacy-utility trade-off by fixing certain weight parameters during training.

There are some improvements the authors could make:
1. Unexplained Discrepancy in Dataset Label Proportions:
One noticeable issue in Table 1 is the difference in the proportion of positive (anomalous) samples between the training set and test set for the ECG dataset (20% in the training set vs. 3.2% in the test set). The paper does not provide a clear explanation for this discrepancy.

2. Limited Dataset Generalization:
The method is only evaluated on ECG and EEG datasets, which are highly specific to the medical domain. The paper would benefit from testing the method on a broader range of sequential data, such as financial or sensor data, to demonstrate the versatility and robustness of the DP-SSSD approach.

3. Lack of Practical Guidelines for Choosing Privacy Parameters:
The paper effectively shows the impact of different privacy levels (𝜖=100,10,1ϵ=100,10,1) on data quality, but it does not provide practical guidelines or recommendations for choosing appropriate privacy parameters in real-world settings. Since selecting the right balance between privacy and utility is crucial for practical applications, offering insights into how to make this choice would enhance the paper’s relevance to practitioners.

---

### Official Review · Reviewer_Ah31 · 2024-10-09
**Differentially Private Sequential Data Synthesis**

**Rating:** 8
**Confidence:** 3

**Review:**

This research paper focuses on generating high-quality synthetic sequential data, such as electrocardiograms (ECGs) and electroencephalograms (EEGs), while preserving the privacy of individuals. The authors propose a new method called Differentially Private Structured State Space Diffusion (DP-SSSD), which combines structured state space models and diffusion models to achieve this goal. They demonstrate that DP-SSSD outperforms previous differentially private sequential data synthesis methods by generating more realistic and diverse data, particularly in the context of ECG and EEG. The authors further explore improving the privacy-utility trade-off by fixing certain weight parameters during training, which yields positive results in certain tasks. The paper concludes by discussing future directions for research, including the development of more compact and noise-resistant sequential generative models to further enhance the quality of synthetic data in strong privacy-preserving settings.